# Autotetraploidization Alters Morphology, Photosynthesis, Cytological Characteristics and Fruit Quality in Sour Jujube (*Ziziphus acidojujuba* Cheng et Liu)

**DOI:** 10.3390/plants12051106

**Published:** 2023-03-01

**Authors:** Lihu Wang, Lixin Wang, Tingting Ye, Jin Zhao, Lili Wang, Hairong Wei, Ping Liu, Mengjun Liu

**Affiliations:** 1Research Center of Chinese Jujube, Hebei Agricultural University, Baoding 071001, China; 2School of Landscape and Ecological Engineering, Hebei University of Engineering, Handan 056038, China; 3Research Institute of Jujube Industry Technology of Hebei, Baoding 071001, China; 4College of Life Sciences, Hebei Agricultural University, Baoding 071001, China; 5School of Forest Resources and Environmental Science, Michigan Technological University, Houghton, MI 49931, USA

**Keywords:** polyploidization, autotetraploid, characteristic, fruit quality

## Abstract

Artificially induced polyploidization is one of the most effective techniques for improving the biological properties and creating new cultivars of fruit trees. Up to now, systematic research on the autotetraploid of sour jujube (*Ziziphus acidojujuba* Cheng et Liu) has not been reported. ‘Zhuguang’ is the first released autotetraploid sour jujube induced with colchicine. The objective of this study was to compare the differences in the morphological, cytological characteristics, and fruit quality between diploid and autotetraploid. Compared with the original diploid, ‘Zhuguang’ showed dwarf phenotypes and decreased tree vigor. The sizes of the flowers, pollen, stomata, and leaves of ‘Zhuguang’ were larger. Perceptible darker green leaves were observed in ‘Zhuguang’ trees owing to increased chlorophyll contents, which led to higher photosynthesis efficiency and bigger fruit. The pollen activities and the contents of ascorbic acid, titratable acid, and soluble sugar in the autotetraploid were lower than those in diploids. However, the cyclic adenosine monophosphate content in autotetraploid fruit was significantly higher. The sugar/acid ratio of autotetraploid fruit was higher than that of diploid fruit, which made the autotetraploid fruit taste different and better. The results indicated that the autotetraploid we generated in sour jujube could greatly meet the goals of our multi-objective optimized breeding strategies for improving sour jujube, which includes tree dwarfing, increased photosynthesis efficiency, and better nutrient values and flavors as well as more bioactive compounds. Needless to say, the autotetraploid can also serve as material for generating valuable triploids or other types of polyploids and are also instrumental in studying the evolution of both sour jujube and Chinese jujube (*Ziziphus jujuba* Mill.).

## 1. Introduction

Polyploidization is one of the main driving forces behind the evolution of plant species [1,2,3]. Previous reports have indicated that most plants have undergone whole-genome duplication during evolution [4,5,6]. The evolutionary pathway of polyploids can be roughly divided into three stages [7]. In the first stage, a plant genome produces a doubling effect under natural stress or artificial intervention. Spontaneous doubling is rare in natural populations. Therefore, polyploid plants are obtained mainly through artificial synthesis at this stage, where the research is mainly focused on the methods and efficiency of mutagenesis. In the second stage, genomic doubling leads to genetic and epigenetic changes, promoting structure and function reorganization. In this stage, polyploid plants generally exhibit superior agronomic characteristics compared to original diploids, such as larger fruit size [8,9], better fruit taste [10], higher nutritional and bioactive component contents [8,11,12,13,14,15], and higher resistance to biotic and abiotic stress [15,16,17,18]. Due to their advantages, polyploid plants, such as tetraploids, are frequently used directly for commercial production or as breeding materials to produce sterile or seedless triploids with many superior or expected advantages [19]. In the third stage, polyploid plants that undergo diploidization with neofunctionalization can eventually produce new species through thousands to hundreds of thousands of years of evolution.

Sour jujube (*Ziziphus acidojujuba* Cheng et Liu) is an economically important fruit shrub or tree species native to China, and it is frequently used as rootstock for grafting Chinese jujube (*Ziziphus jujuba* Mill.) [20]. The fruits of sour jujube are of high economic and medicinal value [21,22]. At the same time, the kernels of sour jujube have been used in traditional herbal medicine for thousands of years to nourish the heart and liver and soothe nerves [23,24,25]. Previous studies have found that ascorbic acid, flavonoid, polysaccharide, and triterpenic acid are the main bioactive components in sour jujube fruit and kernels [26]. Sour jujube is regarded as the direct ancestor of the Chinese jujube. Thus, studying the polyploidy of sour jujube may shed some light on the evolution of Chinese jujube’s characteristics.

The genetic selection of sour jujube germplasms was initiated in the 1980s [20,27]. Up to now, the spontaneous polyploidy of sour jujube under natural conditions has not been found. The artificial polyploidization of sour jujube was started in 2005 [28]. After years of scrutiny, our research team designed a method for the rapid in vivo induction of homogeneous autopolyploids from the callus tissue of sour jujube [29] and successively obtained autotetraploid and octoploid sour jujube [27,30,31]. However, previous studies have mainly focused on developing polyploid induction methods and increasing induction efficiency. The assessment of morphological growth, phenotypic characteristics, and the fruit nutrients of autotetraploid in the field has not been reported.

In this study, the morphological, cytological, and fruit quality changes of autotetraploid ‘Zhuguang’ were comparatively studied in comparison with its diploid ‘Xingtai 0604′. The results provide supportive evidence for the feasibility of improving sour jujube by generating an autotetraploid, meeting the goals of our multi-objective breeding strategy for sour jujube. The autotetraploid generated can serve as material for the generation of valuable triploids or other types of polyploids and are also instrumental in studying the evolution of both sour and Chinese jujube.

## 2. Materials and Methods

### 2.1. Plant Material and Cultivating Conditions

Sour jujube cultivars of ‘Zhuguang’ (autotetraploid) [32] and ‘Xingtai 0604′ (diploid) were grafted onto 6-year-old ‘Zanhuangdazao’ trees in April 2012, which were propagated by suckers. Three mean trees representative of each population were selected to evaluate the diploid and autotetraploid in 2016. All of the sample trees were routinely managed in Zanhuang county, Hebei province, at 37°67′ north latitude and 114°31′ east longitude, with an average annual temperature of 13.3 °C and an annual precipitation of 568 mm.

### 2.2. Examination of Ploidy by Chromosome Counting

To examine the chromosomal numbers in the autotetraploid, we performed karyotype analysis on the shoot tips of the ‘Xingtai 0604′ and ‘Zhuguang’ trees. The shoot tips were collected at 9 am in the morning in mid-May and pretreated with 0.02 M of 8-hydroxyquinoline for 2 h. After pretreatment, the shoot tips were transferred to 0.075 M KCl for 0.5 h at 25 °C, then transferred to Carnoy’s fixative solution (ethanol: glacial acetic acid = 3:1) to fix for 24 h at 4 °C. The shoot tips were then rinsed with distilled water 6 times and then macerated for 6 h with a 2.5% mixed enzyme solution (pectinase: cellulase = 1:1). The enzyme solution was gently sucked with a dropper, washed with distilled water 3 times, and then infused with distilled water for 60 min. A drop of fixative was added to the center of a pre-frozen clean slide after these shoot tips were quickly mashed into the fixative with tweezers. The slide was placed on an alcohol lamp and heated until dry. The dried slides were stained with 5% Giemsa solution for 1 h at 25 °C. The chromosome slides were obtained after the slides were washed with tap water several times and then dried. A photomicroscope (Olympus BX41, Japan) was used for chromosome determination. Images were taken with Image Analysis System 10.0 software.

### 2.3. Evaluation of Tree Characteristics

Multiple characteristics of the ‘Zhuguang’ variety, including but not limited to the posture, form, vigor, plant height, annual shoot growth, and the annual number of extension shoots, were measured. Tree posture, form, and vigor were assessed using a method described by Li [33]. To measure the annual growth of the shoots, nine 5-year-old branches with the same growth status were selected from both the ‘Zhuguang’ and ‘Xingtai 0604′ varieties, and the lengths of the 1-year-old extension shoots were measured. To determine the number of extension shoots, three trees were selected from the diploids and autotetraploid, and the number of extension shoots in one year was checked and recorded.

### 2.4. Evaluation of Flower Characteristics

In order to clarify the change in flower characteristics after chromosome doubling. Thirty flower buds and flowers were randomly selected from ‘Zhuguang’ and ‘Xingtai 0604′, and the flower bud diameters, the diameters of the flowers in the full-bloom stage, the number of stamens, sepal lengths, petal lengths, petal widths, anther lengths, and anther widths were investigated, and the mean values were then calculated. The mean diameter of the pollen grains was determined by following methods described earlier [30].

Ten flower buds were randomly selected from both ‘Zhuguang’ and ‘Xingtai 0604′ during the full-bloom stage for the in vitro pollen germination test; the pollens were spread on solid culture media (0.01% boric acid, 15% sucrose and 0.5% agar) at 26 °C in the dark for 24 h. To determine the pollen germination rates, 10 Petri dishes for either ‘Zhuguang’ or ‘Xingtai 0604′ were observed under a photomicroscope (Olympus BX41, Tokyo, Japan) and counted. Images were taken using Image Analysis System 10.0 software.

### 2.5. Evaluation of Leaf Characteristics

Some leaf characteristics were determined and compared between ‘Zhuguang’ and ‘Xingtai 0604′, including the widths, lengths, colors, epidermal cell structure, chlorophyll contents, and photosynthetic parameters. Thirty leaves were selected from ‘Zhuguang’ and ‘Xingtai 0604′ and were used to determine leaf length, width, and color. The epidermal cell structure was investigated and calculated following the methods of Shi et al. [30,31]. The chlorophyll contents were determined using the method used in a previous study [34]. The photosynthetic parameters were measured using a multi-leaf chamber dynamic photosynthetic apparatus (YZQ-100E, Yicongqi, Beijing, China) every 2 h from 6:00 am to 18:00 pm, including the net photosynthetic rate (Pn), internal CO_2_ concentration (Ci), stomatal conductance (Gs), and transpiration rate (Tr).

### 2.6. Evaluation of Fruit Quality

Thirty mature fruits were selected from both ‘Xingtai 0604′ and ‘Zhuguang’, which were then used to determine fruit quality, including vertical diameter, cross-diameter, shape index, and weight and the contents of ascorbic acid (Vc), titratable acid, soluble sugar, and cyclic adenosine monophosphate (cAMP). The content of ascorbic acid (Vc), titratable acid, soluble sugar, and cyclic adenosine monophosphate (cAMP) were determined using the methods of previous studies [35,36,37].

### 2.7. Statistical Analysis

The data were analyzed using R statistical software (R Development, Core Team, 2018). Student’s t-test was used to evaluate the statistical significance. The figures were made using an R package called ggplot2 in the R statistical software.

## 3. Results

### 3.1. Karyotypic Evidence Showed ‘Zhuguang’ Was Autotetraploid

Karyotype analysis revealed that ‘Zhuguang’ was indeed an autotetraploid, with 48 chromosomes (Figure 1B), whereas ‘Xingtai 0604′ had 24 chromosomes (Figure 1A). This result indicated that ‘Zhuguang’ has maintained its stable ploidy level for many years and can represent an autotetraploid in our evaluation.

### 3.2. Evaluation of Tree Characteristics

The autotetraploid ‘Zhuguang’ plant displayed distinct morphological characteristics compared to the diploid plant (Figure 2). Compared with ‘Xingtai0604′, Zhuguang’ lacked apical dominance and thus had a more opened canopy. Autotetraploid plants exhibit a certain dwarf effect compared with diploid plants, and the data showed that the plant height of the autotetraploid was 21.54% lower than those of the diploid.

At the same time, the growth rate of ‘Zhuguang’ is significantly lower than that of ‘Xingtai0604′. The result showed that the annual growth of shoots and the annual number of extension shoots of ‘Xingtai0604′ were higher than ‘Zhuguang’ by 28.29% and 135.34%, respectively. Compared with ‘Xingtai0604′, ‘Zhuguang’ showed poor dryness (Table 1, Figure 2A–D) and slower tree growth. All of these data indicated that the ‘Zhuguang’ plants exhibited a reduced tree vigor compared to the ‘Xingtai0604′ plants.

### 3.3. Evaluation of Flower Characteristics

The flower characteristics of ‘Xingtai0604′ and ‘Zhuguang’, including the average diameter of the flower buds and blooming flowers, number of stamens, sepal length, petal length, petal width, anther length, anther width, and the diameter of their pollen grains, were investigated. By comparing the ‘Zhuguang’ and ‘Xingtai0604′ flower characteristics, as shown in Figure 3 and Table 2, we found that the flowers of ‘Zhuguang’ were much larger than those of ‘Xingtai0604′.

The average diameter of the flower buds and flowers in full bloom, petal length, sepal length, sepal width, anther length, anther width, and the diameter of the pollen grains of ‘Zhuguang’ increased by 25.69%, 26.59%, 18.10%, 13.01%, 24.58%, 36.36%, and 11.43% compared to those of ‘Xingtai0604′, respectively, as shown in Table 2.

As shown in Figure 4 and Table 2, we found that the pollens of ‘Zhuguang’ were much larger than those of ‘Xingtai 0604′. The average pollen diameter of ‘Zhuguang’ was 22.09% larger than that of diploids. In addition, the pollen activity of ‘Zhuguang’ significantly decreased compared to Xingtai 0604, with the germination rate being usually 27.21% (Figure 4C,D), much less than the 53.01% of ‘Xingtai 0604′.

### 3.4. Evaluation of Leaf Characteristics

The morphological characteristics of the autotetraploid ‘Zhuguang’ sour jujube leaves and the lengths of the bearing shoots differed significantly from those of their diploid counterparts (Figure 5). The lengths of the bearing shoots of ‘Zhuguang’ increased by 9.58%, but the number of leaves on the bearing shoots did not differ significantly in terms of ploidy level.

The leaves of ‘Zhuguang’ were much larger and slightly curled compared to those of ‘Xingtai 0604′. The average length and width of the ‘Zhuguang’ leaves were 7.85% and 56.56% higher than those of ‘Xingtai 0604′, respectively. Significantly enlarged leaves have been observed in almost all polyploids, including autotetraploids, primarily due to ploidy-dependent cell enlargement caused by the increased gene copies [38], leading to the production of larger amounts of proteins and, subsequently, a larger amount of carbohydrates. In addition to leaf size, the color of the ‘Zhuguang’ leaves was obviously darker than those of ‘Xingtai 0604′ leaves, indicating augmented chlorophyll biosynthesis.

The contents of chlorophyll a, chlorophyll b, and total chlorophyll were detected and calculated. As shown in Table 3, the contents of chlorophyll a, chlorophyll b, and the total chlorophyll of ‘Zhuguang’ significantly increased by 22.43%, 58.06%, and 30.43% compared to those of ‘Xingtai 0604′, respectively. Of these, chlorophyll b, which functions in photosynthesis by absorbing light energy [39], increased the most.

The anatomical study of the leaves of both ‘Xingtai 0604′ and autotetraploid leaves revealed that the autotetraploid plants exhibited much larger stomata than the diploid plants (Figure 6A,B). As shown in Table 3, the average width and length of the ‘Zhuguang’ stomata were 31.03% and 30.26% greater than those of ‘Xingtai 0604′, respectively. However, the average stomatal density of ‘Xingtai 0604′ was significantly higher than that of ‘Zhuguang’, which is presumably caused by ploidy-dependent cell enlargement, causing the stomatal apertures to grow larger.

The significantly increased contents of chlorophyll a, chlorophyll b, and the total chlorophyll and the enlarged stomatal apertures should have positive impacts on photosynthesis. Thus, we measured the photosynthesis of ‘Zhuguang’. As shown in Figure 7, the photosynthetic rate, transpiration rate, stomatal conductance, and internal CO2 concentration of ‘Zhuguang’ were significantly higher than those of diploids. However, the diurnal photosynthetic changes in the photosynthetic rate, transpiration rate, and stomatal conductance of ‘Zhuguang’ had a similar trend. The maximum values of the photosynthetic rate, transpiration rate, and stomatal conductance in ‘Zhuguang’ were increased by 121.35%, 141.53% and 206.92%, respectively. The diurnal change trend of internal CO2 concentration was similar, and there was no significant difference between ‘Xingtai 0604′ and ‘Zhuguang’.

### 3.5. Autotetraploid Fruit Showed Changed of Quality

A significant difference in the fruit sizes of ‘Xingtai 0604′ and ‘Zhuguang’ was observed (Figure 8). As shown in Table 4, the fruit length and width of ‘Zhuguang’ were, on average, 17.45%, and 20.56% larger than those of ‘Xingtai 0604′, respectively. The fruit index of ‘Xingtai 0604′ was higher than ‘Zhuguang’. The data indicated that the change in fruit width exceeded the change in fruit length after the homologous doubling of diploid sour jujube. The fruit shape index, which represents the ratio of fruit length to width, decreased.

The analysis of the nutrients in the fruit suggests that the ‘Zhuguang’ fruit showed improved quality. For example, the cAMP content of ‘Zhuguang’ was 38% higher than that of ‘Xingtai 0604′. On the other hand, Vc, titratable acid, and the soluble sugar contents of ‘Zhuguang’ decreased by 51.43%, 16.66%, and 12.72%, respectively. It is worth noting that the sugar–acid ratio of ‘Zhuguang’ was higher than the diploid ‘Xingtai 0604′.

## 4. Discussion

Polyploidization is one of the most effective approaches to improving fruit quality. In the previous studies, autotetraploid plants were shown to bear much larger fruit; for example, autotetraploid trees of apple [40] and banana [38], and autotetraploid plants of muskmelon [8,41]. On the contrary, the average fruit size of autotetraploid pears is approximately the same as that of diploids, though the autotetraploid fruit grows quicker than diploids during the early stages [10]. In this study, the average size of autotetraploid sour jujube ‘Zhuguang’ fruit was significantly larger than that of diploids. As the important indicators of the flavor and nutritional quality of fruits, the sugar and acid contents of the plants change in response to chromosome doubling. The sugar content of the autotetraploid fruit was reported to be significantly higher than that of the diploids [8,10]. However, in this study, the sugar and acid contents of autotetraploid sour jujube fruit were lower than those of diploids, though the ratio of sugar to acid was higher, which can change the fruit flavor. Reduced acid content has also been reported in other autotetraploid fruit trees [10,12]. Therefore, the changes in fruit sizes and the sugar and acid contents of the plants after tetraploidization depend on the species. Although the Vc content of ‘Zhuguang’ was reduced, it is still significantly higher than that of other types of fruit, such as raspberries, blackberries, red currants, gooseberries, and cornelian cherries [42]. At the same time, the cAMP content of ‘Zhuguang’ has also been improved compared to that in the diploid. In summary, the fruit of ‘Zhuguang’ still has excellent fruit quality and may be more suitable for use by hyperglycemic people.

Sour jujube is considered to be the wild ancestor of Chinese jujube, and many studies have shown some evidence that Chinese jujube evolved from sour jujube [20]. Most Chinese jujube cultivars have prototypes in sour jujube [43]. Compared with sour jujube, Chinese jujube exhibits larger fruit, higher sugar content, lower acid content, lower Vc content, and a higher cAMP content [35,44]. The molecular explanation for the difference in sugar and acid content between Chinese jujube and sour jujube is that the acid biosynthesis genes are highly expressed, and the sugar biosynthesis genes are lowly expressed in sour jujube [45] compared to those in Chinese jujube. In this paper, autotetraploid sour jujube ‘Zhuguang’ showed similar nutritional characteristics as Chinese jujube. Currently, we are still not sure if Chinese jujube evolved these characteristics through neofunctionalization upon a genome duplication. Genomic sequencing data demonstrate that the Chinese jujube genome has undergone frequent inter-chromosome fusions and segmental duplications, but there was a lack of a recent whole-genome duplication after gamma duplication [46]. In addition, whether the whole-genome duplication event has occurred in the evolution of the sour jujube genome is also unknown because the whole genome sequencing of sour jujube has not yet been completed. Additionally, the natural polyploidy of sour jujube has not been found in nature.

Autotetraploid plants are usually used as intermediate materials to obtain triploids with outstanding agronomic characteristics, such as disease resistance, larger fruit size, and an absence of seeds [38]. One barrier to use an autotetraploid as a paternal parent for crossing is that the pollens of an autotetraploid have lower activity. Previous studies have demonstrated that the mitosis of autotetraploid pollens is abnormal and often results in reduced pollen activity and lower fertility rates, which can eventually affect fruit settings, seed numbers, and the seed germination rates of the hybrid fruits [19,47,48,49]. Although autotetraploid pollen activity and fertility were reduced, it did not significantly affect the acquisition of triploid seeds via crossing. The results of previous studies also indicated that genome multiplication mainly affected pollen activity rather than the fertility of the female reproductive organs [10]. In our previous study, triploids of Chinese jujube were obtained successfully using autotetraploid Chinese jujube as the male parent and diploid Chinese jujube as the female parent [50]. As an ancestor of Chinese jujube, sour jujube has adapted well to harsher environmental conditions compared to Chinese jujube. In this study, we found that autotetraploid sour jujube can produce pollens that have about a 30% germination rate, and autotetraploid themselves bear no seeds due to ovary abortion [47,51], which makes autotetraploid sour jujube ideal for use as a paternal parent to obtain triploid hybrids of sour jujube (2n) and sour jujube (4n) or the triploid hybrids of Chinese jujube (2n) and sour jujube (4n) for augmented adaptation and enlarged fruit. The autotetraploid can be used as starting material to generate octoploids, which may have more desirable traits.

## 5. Conclusions

The autotetraploid sour jujube ‘Zhuguang’ exhibited excellent agronomic characteristics that have been sought after. These include dwarf phenotypes, opened canopies, enlarged fruit sizes, and improved flavor. Therefore, ‘Zhuguang’ is a new valuable sour jujube cultivar and an important genetic material for generating other polyploids, for instance, triploids and octoploids, through hybridization or artificial doubling. The autotetraploid can also be used to study the evolution of Chinese jujube from sour jujube.

## Figures and Tables

**Figure 1 plants-12-01106-f001:**
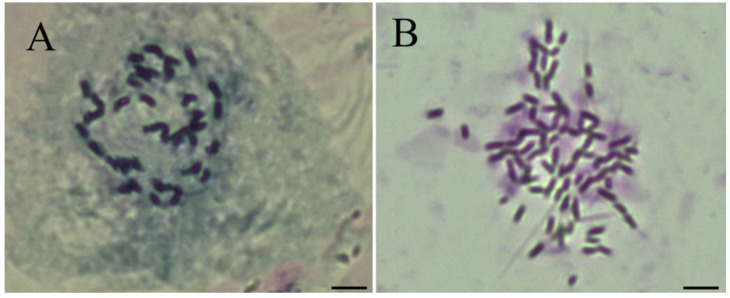
Karyotype analysis of chromosomes of deploid ‘Xingtao 0604′ ((**A**), 2n = 2*x* = 24) and autotetraploid ‘Zhuguang’ ((**B**), 2n = 4*x* = 48), Bars = 5 μm.

**Figure 2 plants-12-01106-f002:**
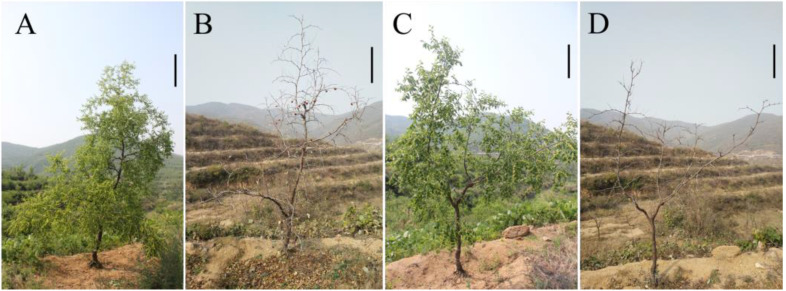
The morphological difference between ‘Xingtai0604′ (diploid) and ‘Zhuguang’ (autotetraploid). (**A**) and (**B**), ‘Xingtai0604′. (**C**) and (**D**), ‘Zhuguang’. Bars = 0.5 m.

**Figure 3 plants-12-01106-f003:**
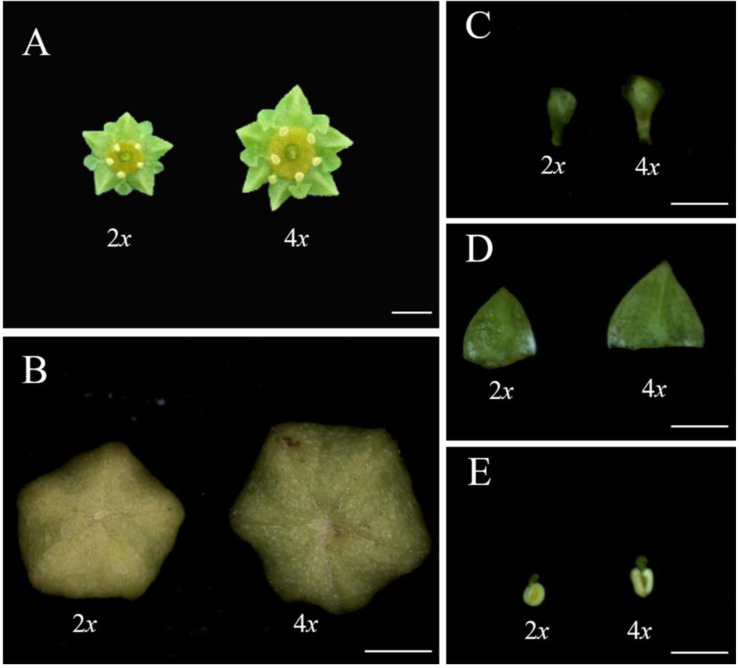
Flower morphologies of ‘Xingtai0604′ (2*x*) and ‘Zhuguang’ (4*x*), (**A**) flowers, bar represents 3 mm. (**B**) flower buds. (**C**) petals. (**D**) sepals. (**E**) anther. The bars in B–E represent 1 mm.

**Figure 4 plants-12-01106-f004:**
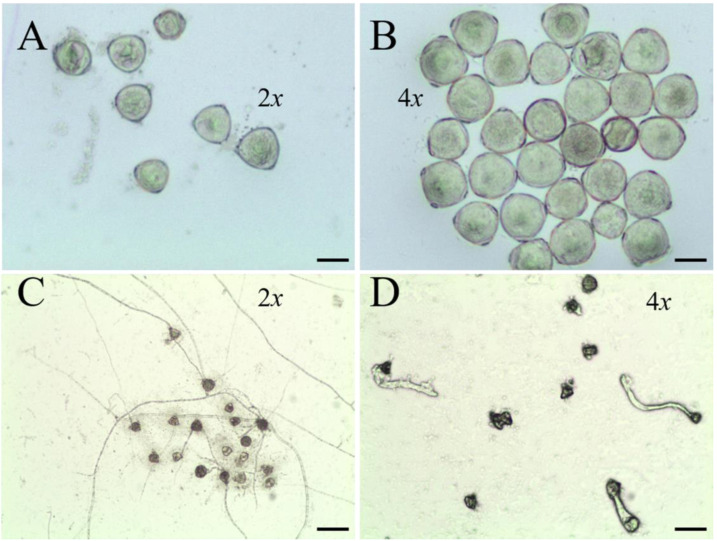
Pollen characteristics of ‘Xingtai 0604′ (2*x*) and ‘Zhuguang’ (4*x*). (**A**) The pollen sizes of ‘Xingtai 0604′. (**B**) the pollen sizes of ‘Zhuguang’. (**C**) the pollen activity of ‘Xingtai 0604′. (**D**) the pollen activity of ‘Zhuguang’, (**A**,**B**), bars = 25 μm, (**C**,**D**), bars = 75 μm.

**Figure 5 plants-12-01106-f005:**
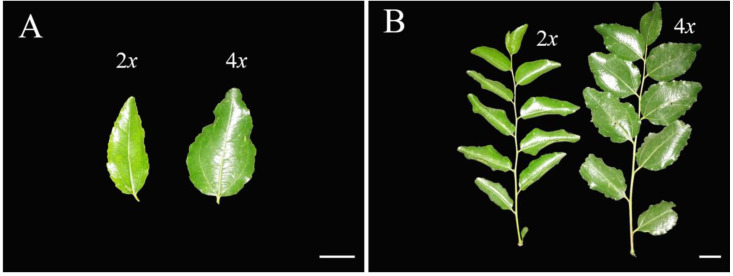
Morphological changes of ‘Xingtai 0604′ (2*x*) and ‘Zhuguang’ (4*x*) leaves and bearing shoot. (**A**): leaves; (**B**): bearing shoot, bars = 2 cm.

**Figure 6 plants-12-01106-f006:**
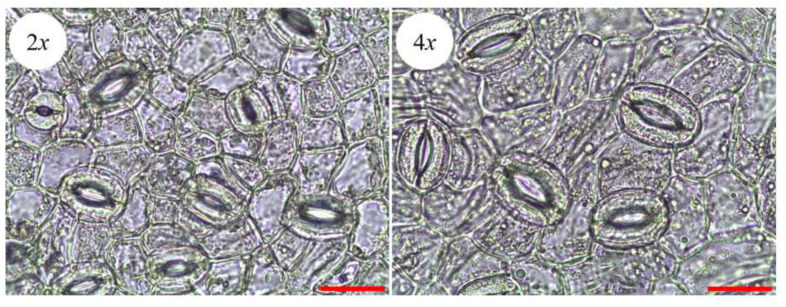
Stoma morphological of ‘Xingtai 0604′ (2*x*) and ‘Zhuguang’ (4*x*). Both bars represent 25 μm.

**Figure 7 plants-12-01106-f007:**
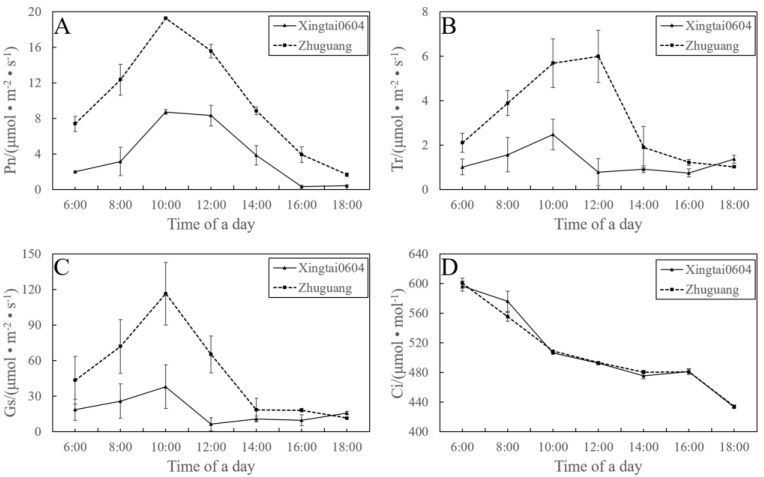
Diurnal photosynthetic parameter changes in ‘Xingtai 0604′ (2*x*) and ‘Zhuguang’ (4*x*). Pn, photosynthetic rate (**A**); Tr, transpiration rate (**B**); Gs, stomatal conductance (**C**); Ci, internal CO_2_ concentration (**D**). Vertical bar at each point represents the standard deviation.

**Figure 8 plants-12-01106-f008:**
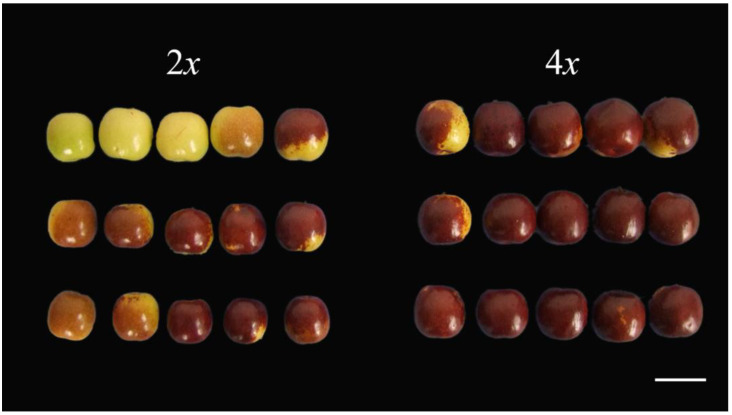
Fruit morphology of ‘Xingtai 0604′ (2*x*) and ‘Zhuguang’ (4*x*), bar = 30 mm.

**Table 1 plants-12-01106-t001:** Tree characteristics of ‘Xingtai 0604′ and ‘Zhuguang’.

Tree Characteristics	Xingtai 0604 (2*x*) ((Mean ± SD)	Zhuguang (4*x*) (Mean ± SD)
Apical dominance	Obvious	Not obvious
Tree form	Circular cone shape	Globose shape
Tree vigor	Vigorous	Intermediate
Plant height (m)	2.99 ± 0.12 *	2.46 ± 0.09
Annual growth of extension shoots (cm)	51.93 ± 7.93 *	40.48 ± 6.56
Annual number of extension shoots	13.33 ± 1.53 *	5.67 ± 2.51

* represents the statistically significant differences between the ‘Xingtai 0604′ and ‘Zhuguang’ at *p* < 0.05 and *p* < 0.01 (Student’s *t*-test), respectively. SD, standard deviation.

**Table 2 plants-12-01106-t002:** Flower characteristics of ‘Xingtai0604′ and ‘Zhuguang’.

Flower Characteristics	Xingtai 0604 (2*x*) (Mean ± SD)	Zhuguang (4*x*) (Mean ± SD)
Flower diameter (mm)	2.88 ± 0.13	3.62 ± 0.24 **
Flower bud diameter (mm)	1.88 ± 0.05	2.38 ± 0.08 **
Petal length (mm)	1.05 ± 0.01	1.24 ± 0.03 **
Sepal length (mm)	1.46 ± 0.03	1.65 ± 0.02 **
Sepal width (mm)	1.18 ± 0.02	1.47 ± 0.04 **
Anther length (mm)	0.55 ± 0.02	0.75 ± 0.02 **
Anther width (mm)	0.35 ± 0.02	0.39 ± 0.03 **
Pollen diameter (μm)	25.67 ± 2.34	31.34 ± 3.07 **
Pollen germination rate (%)	53.01 ± 6.06 **	27.21 ± 7.20

** represents the statistically significant differences between ‘Xingtai 0604′ and ‘Zhuguang’ at *p* < 0.05 and *p* < 0.01 (Student’s *t*-test), respectively. SD, standard deviation.

**Table 3 plants-12-01106-t003:** Leaf characteristics of ‘Xingtai 0604′ and ‘Zhuguang’.

Leaf Characteristics	Xingtai 0604 (2*x*) (Mean ± SD)	Zhuguang (4*x*) (Mean ± SD)
Length of bearing shoots (cm)	14.62 ± 1.60	16.02 ± 1.35 *
Number of leaves on fruit-bearing shoot	10	10
Leaf length (cm)	4.20 ± 0.42	4.53 ± 0.47 *
Leaf width (cm)	2.21 ± 0.28	3.46 ± 0.27 *
Stoma length (μm)	26.68 ± 1.08	34.96 ± 1.17 **
Stoma width (μm)	18.54 ± 0.69	24.15 ± 0.95 **
Stomatal density (mm^−2^)	275.87 ± 8.66 **	162.67 ± 8.53
Chlorophyll a (mg/g)	2.14 ± 0.12	2.62 ± 0.06 *
Chlorophyll b (mg/g)	0.62 ± 0.01	0.98 ± 0.05 **
Chlorophyll (mg/g)	2.76 ± 0.12	3.60 ± 0.08 **

* and ** represent the statistical significance differences between ‘Xingtai 0604′ and ‘Zhuguang’ at *p* < 0.05 and *p* < 0.01 (Student’s t-test), respectively. SD, standard deviation.

**Table 4 plants-12-01106-t004:** Fruit characteristics and quality of ‘Xingtai 0604′ and ‘Zhuguang’.

Fruit Quality	Xingtai 0604 (2*x*) (Mean ± SD)	Zhuguang (4*x*) (Mean ± SD)
Fruit length (mm)	24.36 ± 0.60	28.61 ± 0.65 **
Fruit width (mm)	25.39 ± 0.77	30.61 ± 1.44 **
Fruit index (length/width)	0.96	0.93
Fruit weight (g)	5.99 ± 1.37	8.03 ± 0.78 **
Vc content (mg/g)	2.65 ± 0.06 **	1.75 ± 0.02
cAMP content (mg/100g)	11.21 ± 0.57	15.47 ± 0.25 *
Titratable acid content (%)	0.49 ± 0.06	0.42 ± 0.02
Soluble sugar content (%)	25.78 ± 0.15 **	22.87 ± 0.11
Soluble sugar/titratable acid	52.61	54.45

* and ** represent the statistical significance differences between ‘Xingtai 0604′ and ‘Zhuguang’ at *p* < 0.05, and *p* < 0.01 (Student’s t-test), respectively. SD, standard deviation.

## Data Availability

Data will be made available upon request.

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
