# Peer review of "Autotetraploidization Alters Morphology, Photosynthesis, Cytological Characteristics and Fruit Quality in Sour Jujube (Ziziphus acidojujuba Cheng et Liu)"

_plants, 2023, doi:10.3390/plants12051106_

Round 1
Reviewer 1 Report
Comments:
(1) Line 24-25, it is better to describe it as ‘The pollen activities and the contents of ascorbic acid, titratable acid, and soluble sugar in autotetraploid were lower than those in diploids’.
(2) Line 134, the title of 3.1 could be changed to ‘Karyotypic evidence showed ‘Zhuguang’ was autotetraploid’.
(3) In the legends of Figure 2, the description like ‘Xingtai0604’, which showed obvious apical dominance. C and D, ‘Zhuguang’, which was lack of apical dominance and thus had a more opened canopy.’ should be delete and put into the result part.
(4) The table 1 should be cited in the result, it could be cited at the end of line 151ï¼›
(5) In figure 6,‘represents’ should be ‘represent’.
(6) Line 223, CO2 should be CO2.
Author Response
Feb 5, 2023
Dear reviewer,
Thank you for reviewer’s kind and instructive comments. We have tried to correct the manuscript according to these comments, and the changes are used the "track changes feature" of document in our modified manuscript.
The manuscript entitled as “Autotetraploidization alters morphology, photosynthesis, cytological characteristics and fruit quality in sour jujube” described the differences of morphological, cytological characteristics, and nutrition in fruits between diploid and its autotetraploid ‘Zhuguang’ which could provide valuable information for ploidy breeding in sour jujube. The work was interesting, after several minor revisions and it could be considered to be published.
(1) Line 24-25, it is better to describe it as ‘The pollen activities and the contents of ascorbic acid, titratable acid, and soluble sugar in autotetraploid were lower than those in diploids’.
< Response > Thanks very much for your suggestions. It was modified in revised manuscript.
(2) Line 134, the title of 3.1 could be changed to ‘Karyotypic evidence showed ‘Zhuguang’ was autotetraploid’.
< Response > Thanks very much for your suggestions. It was modified in revised manuscript.
(3) In the legends of Figure 2, the description like ‘Xingtai0604’, which showed obvious apical dominance. C and D, ‘Zhuguang’, which was lack of apical dominance and thus had a more opened canopy.’ should be delete and put into the result part.
< Response > Thanks very much for your suggestions. It was modified in revised manuscript.
(4) The table 1 should be cited in the result, it could be cited at the end of line 151ï¼›
< Response > Thanks very much for your suggestions. The table 1 have been cited in the result, It was modified in revised manuscript.
(5) In figure 6,‘represents’ should be ‘represent’.
< Response >It was modified in revised manuscript. we have changed ‘represents’ into ‘represent’.
(6) Line 223, CO2 should be CO2.
< Response >I'm very sorry for this mistake. It was modified in revised manuscript.
In addition, we have re-check the full manuscript and improved it.
We hope our revision could meet your request.
Thank you again!
Mengjun Liu
Research Center of Chinese Jujube
Hebei Agricultural University
289, Lingyushi Street
Baoding,071001, China
Phone: +8613932262298
Email: lmj1234567@aliyun.com

Reviewer 2 Report
The manuscript describes the comparison of some morphological, physiological, and horticultural traits between a tetraploid sour jujube and its ancestry diploid cultivar. Although polyploidy induced changes in botanic and biological traits are frequently reported, there is no detailed information in sour jujube. Sufficient data are presented in the manuscript to elucidate the differences between the two cultivars with different chromosome ploidy. I thus suggest acceptance for publication after minor revision.
Line 1, the scientific name of sour jujube should be given in the title because this species is not popularly known for common readers. Also, the scientific names of sour and Chinese jujube should be given again in the abstract.
Line 51, the genus name Z. should be spelled out when the first appearance.
Line 52 the scientific name of Chinese jujube should be given when the first appearance in the text.
Line 52-55, 56-57, citations are needed.
Line 75, are the plant materials sour jujube? it must be stated clearly. In addition, the detailed information about the origin of 'Zhuguang' must be presented at MM section, or relevant references be cited.
Line 80-93, the description is actually a protocol for shoot tip chromosome counting, other than for karyotype analysis.
Line 228, the subheading does not match the result because the contents of soluble sugar and ascorbic acid were significantly lower in 'Zhuguang' than in the diploid ancestor.
Line 244-266, Why the authors discuss the use of tetraploid 'Zhuguang' as rootstock of Chinese jujube? They present no data to support. My suggestion is that the paragraph be deleted completely.
Line 314, "of" should be deleted.
Line 334-502, the format of reference list needs to be edited, ie the abbreviation or not of the journal names, the capitalization of words in titles, list of all authors or using et al. etc.
Line 403, spelling check "jujubein".
Author Response
Feb 5, 2023
Dear reviewer
Thank you for reviewer’s kind and instructive comments. We have tried to correct the manuscript according to these comments, and the changes are used the "track changes feature" of document in our modified manuscript.
The manuscript describes the comparison of some morphological, physiological, and horticultural traits between a tetraploid sour jujube and its ancestry diploid cultivar. Although polyploidy induced changes in botanic and biological traits are frequently reported, there is no detailed information in sour jujube. Sufficient data are presented in the manuscript to elucidate the differences between the two cultivars with different chromosome ploidy. I thus suggest acceptance for publication after minor revision.
Line 1, the scientific name of sour jujube should be given in the title because this species is not popularly known for common readers. Also, the scientific names of sour and Chinese jujube should be given again in the abstract.
< Response >Thanks very much for your suggestions. In the revised manuscript, We made the appropriate changes based on your suggestions.
Line 51, the genus name Z. should be spelled out when the first appearance.
< Response >Thanks very much for your suggestions. In the revised manuscript, We made the appropriate changes based on your suggestions.
Line 52 the scientific name of Chinese jujube should be given when the first appearance in the text.
< Response >Thanks very much for your suggestions. In the revised manuscript, We made the appropriate changes based on your suggestions.
Line 52-55, 56-57, citations are needed.
< Response >Thanks very much for your suggestions. It was modified in revised manuscript.
Line 75, are the plant materials sour jujube? it must be stated clearly. In addition, the detailed information about the origin of 'Zhuguang' must be presented at MM section, or relevant references be cited.
< Response >Thank you for your suggestion. we made it clear that the material was sour jujube and relevant literature was cited in revised manuscript.
Line 80-93, the description is actually a protocol for shoot tip chromosome counting, other than for karyotype analysis.
< Response > Thanks very much for your suggestions. It was modified in revised manuscript.
Line 228, the subheading does not match the result because the contents of soluble sugar and ascorbic acid were significantly lower in 'Zhuguang' than in the diploid ancestor.
< Response >It was modified in revised manuscript.
Line 244-266, Why the authors discuss the use of tetraploid 'Zhuguang' as rootstock of Chinese jujube? They present no data to support. My suggestion is that the paragraph be deleted completely.
< Response >Thanks for your suggestion, we have discussed and deleted this paragraph in revised manuscript.
Line 314, "of" should be deleted.
< Response > The "of" have been deleted in revised manuscript.
Line 334-502, the format of reference list needs to be edited, ie the abbreviation or not of the journal names, the capitalization of words in titles, list of all authors or using et al. etc.
< Response > Thanks to your suggestions, we have revised the format of the references and sorted them using the document management software endnote
Line 403, spelling check "jujubein"
< Response >It was modified in revised manuscript.
In addition, we have re-check the full manuscript and improved it.
We hope our revision could meet your request.
Thank you again!
Mengjun Liu
Research Center of Chinese Jujube
Hebei Agricultural University
289, Lingyushi Street
Baoding,071001, China
Phone: +8613932262298
Email: lmj1234567@aliyun.com
